# A Complexity Science Account of Humor

**DOI:** 10.3390/e25020341

**Published:** 2023-02-13

**Authors:** Wolfgang Tschacher, Hermann Haken

**Affiliations:** 1Department of Experimental Psychology, University Hospital of Psychiatry and Psychotherapy, University of Bern, 3060 Bern, Switzerland; 2Institute of Theoretical Physics, University of Stuttgart, 70174 Stuttgart, Germany

**Keywords:** attractor, cognitive dissonance, free energy, jokes, funny cartoons, phase transition, self-organization, synergetics

## Abstract

A common assumption of psychological theories of humor is that experienced funniness results from an incongruity between stimuli provided by a verbal joke or visual pun, followed by a sudden, surprising resolution of incongruity. In the perspective of complexity science, this characteristic incongruity-resolution sequence is modeled by a phase transition, where an initial attractor-like script, suggested by the initial joke information, is suddenly destructed, and in the course of resolution replaced by a less probable novel script. The transition from the initial to the enforced final script was modeled as a succession of two attractors with different minimum potentials, during which free energy becomes available to the joke recipient. Hypotheses derived from the model were tested in an empirical study where participants rated the funniness of visual puns. It was found, consistent with the model, that the extent of incongruity and the abruptness of resolution were associated with reported funniness, and with social factors, such as disparagement (*Schadenfreude*) added to humor responses. The model suggests explanations as to why bistable puns and phase transitions in conventional problem solving, albeit also based on phase transitions, are generally less funny. We proposed that findings from the model can be transferred to decision processes and mental change dynamics in psychotherapy.

## 1. Introduction

Psychological theories of humor and joking have postulated that experienced funniness is associated with the incongruity contained in a verbal joke, a visual pun, or a cartoon. Eysenck [1] (p. 307) wrote, “... laughter results from the sudden, insightful integration of contradictory or incongruous ideas, attitudes, or sentiments which are experienced objectively”. Thus, the recipient of humorous stimuli is made to resolve the incongruity, which, if successful, entails an emotional response of amusement, hilarity, and mirth [2,3]. Humor is based on conflict and cognitive dissonance between stimuli. Additionally, it relies on an implicit step in perception and does not require explicit cognitive processing—explaining a joke is not funny. We may, therefore, assume that humor processing is linked with gestalt-like wholistic coordination and integration of stimuli, which occurs spontaneously and at an early stage of perception.

Humor is always an ingredient of direct or mediated social interaction between people (and is observed also in some other mammals). In general, humor occurs in a playful, nonserious social context [4]. Above the appreciation of jokes and cartoons, there are more aspects and forms of humor, which may also come as irony, sarcasm, black humor, situational humor, absurdity, ridicule, etc. The social functions of humor are, however, prominent in its friendly, as well as its more aggressive, mocking, and scornful, varieties. Positive affect and joyful emotions are essential for the amused perceiver of humorous stimuli, and the response is behaviorally embodied in laughter that communicates information to others. Interestingly, laughter belongs to those bodily expressions that are most contagious, and thus tend to synchronize people [5], except for those who are being laughed at (in disparagement or *Schadenfreude* humor [6]. *Schadenfreude* is a specific German word for ‘malicious joy’). To get the joke in social and especially disparagement humor, the perceiver must switch between the perspectives of the protagonists in the joke or pun, and thus must possess mentalizing abilities, or in cognitive terminology, theory of mind (ToM).

Responses to funny stimuli entail a range of physiological changes in the autonomic nervous system, especially in its sympathetic branch, such as increased heart rate, heart rate variability, muscle tension (in laughter), and skin conductance in line with the generally activating and arousing quality of humor. Longer humorous stimulation is linked with endocrinal changes, as it increases the concentration of cortisol and adrenaline [4] (chapter 6).

Here comes a joke: ‘A young university graduate arrives for the first working day of his new job. The boss says, “Now take this broom and sweep the office”. The young man is shocked and says, “But I just finished university!” The boss hesitates and replies, “Oh I see, please excuse me! I will show you how to go about it”’.

In this joke, the punchline is “I will show you how to go about it”. The information (stimulus S1) that was provided prior to this last sentence created a Gestalt, script, or mental schema of the new employee mistaken for the cleaning person, him protesting, and successfully (“please excuse me”) correcting the misunderstanding of his new boss. The second information (stimulus S2) induces incongruity as the boss apparently insists on the initial order to clean the office, and completely reverses the meaning of his excuse. As a result, the incongruity is resolved by even aggravating the boss’s derogatory attitude towards the graduated new employee. S2 has destructed the Gestalt induced by S1. The point of the joke is best accessible to a recipient who can take, and switch between, the perspectives of both protagonists in the joke. Psychologically, an additional contribution to funniness in this joke is *Schadenfreude* about conceited holders of university degrees. Incongruity is resolved completely in this joke because the boss’s utterance is in complete accordance with his extreme pessimism related to just graduated staff.

The second example is a so-called Chuck Norris joke (it is helpful to know that Chuck Norris is a famous, in the USA, actor and hero of numerous martial arts and action movies): ‘Chuck Norris was bitten in the leg by a king cobra. After five days of agonizing pain—the cobra died.’

Here, the punchline is “the cobra died”. It is incongruous with the expectation that the person bitten should have been in pain, and thus eventually even die. The resolution of this incongruity is the ridiculously superhuman property of Chuck Norris, who can never be defeated. Yet even though Chuck Norris as protagonist ‘explains’ the fate of the cobra, the incongruity is not completely resolved, and some nonsense or absurdity remains after the punchline. Such incongruity resolution is called ‘incomplete’ [2].

Jokes need not be verbal; in visual puns (i.e., cartoons without captions or speech balloons) the incongruity is provided by certain visual elements that are ambiguous and conform with two different Gestalts (in linguistic terms, scripts) with antagonistic meanings. Thus, the functioning is equivalent in jokes and puns, whereas the neurocognitive modalities involved are different: linguistic in jokes and visual in puns. Visual puns can be ‘perfect’ [7] or ‘imperfect’; they are called perfect when the ambiguous visual element is depicted in a way that is fully compatible with both scripts. Thus, perfect puns are similar to jokes that allow complete resolution. Imperfect puns are the visual analogues to incomplete jokes.

The goal of this article is to model the processes relevant in humor appreciation in a general manner by applying complexity theory to mental systems. We wish to propose a complexity science account of the patterns that arise in the perceiver in the course of listening to a joke or when viewing visual puns. Deriving hypotheses from this model, we tested these using an empirical dataset of human participants, and thus connected our model to psychological concepts. As material we will restrict our theoretical treatment to jokes and puns; the empirical data was collected using visual puns only.

Based on the theories of humor outlined above, we formulated exploratory hypo-theses. Puns showing social situations and human protagonists, and thus requiring mentalizing, and allowing for *Schadenfreude* humor may be perceived funnier; funniness should be related to mentalizing abilities (Hypothesis 1). Prior affective states of participants may support the funniness elicited by puns (Hypothesis 2). We further assumed that humor appreciation may be associated with personality traits (Hypothesis 3) and intelligence (Hypothesis 4).

In Section 3, we will develop the theoretical model of humor appreciation and formulate further hypotheses that accord with this model. Subsequently, in Section 3.2, we will test all hypotheses using the empirical database.

## 2. Materials and Methods

The methods we applied to arrive at a model of humor are first the concepts of synergetics, which is a theory of self-organization processes in complex systems. Synergetics is a transdisciplinary, structural science because it is abstract in that it is not restricted to a specific ontology, and thus may be applied to physical and mental systems. Pattern formation processes can be observed in physical complex systems, such as the brain [8], but in subjective experience as well, for example, in Gestalt perception [9]. The core phenomenon of self-organization is this: from an initial situation (A) of full complexity, where the state of a system must be described by the information of all system components, a new state (B) emerges that is governed by just a few variables, the order parameters [10]. The transition from the initial to the emergent state is characterized by a drastic reduction of complexity and Shannon information, as the new system in B can now be fully described by just one or few order parameters instead of the complete set of all system components. An important attribute of the self-organized pattern is its stability in the face of random perturbations. Accordingly, state B can be described as an attractor, such as a fixed point (Figure 1).

The concept of probability mentioned in Figure 1 is in need of explanation. In a constant gravitational field, as assumed here, all positions of the ‘ball’ are energetically equivalent. We may find it at any position with equal probability. The situation changes when the field is nonuniform, such as when the corresponding potential energy has the form of a parabola. To allow the introduction of the notion of probability we resort to a model of statistical mechanics. Considering an ensemble (a ‘gas’) of particles each representing the ball, we may assume that they move without collisions in the force field corresponding to the parabolic potential. The probability *P*(*x*) of finding a particle at position *x* in interval dx, i.e., *P*(*x*)dx, is defined as the number of particles in that interval divided by their total number. In the present context, it is only important that *P*(*x*) depends on *x* and is highest at *x* = 0. Both in physics and psychology, *P*(*x*) can be measured experimentally.

Self-organization processes occur spontaneously by ‘themselves’, yet there is a prerequisite for such processes: Self-organization demands that the systems are open and situated far from thermal equilibrium (in physical systems) or underlying motivational tensions (in mental systems). Open systems have an environment, and in the case of spontaneous pattern formation, the environment is valent in the sense of providing gradients of energy or, in mental and social systems, semantic information, affect, or motivation [11]. In synergetics, these gradients are called control parameters, which drive the system away from equilibrium (so that the second law of thermodynamics, which demands preservation or increase of entropy in a system, no longer holds). Even though the presence of an activating environment is a necessary precondition for self-organization, the emerging patterns are still spontaneous and *self*-organized because the gradients are unspecific and do not determine the qualitative shape of the emerging patterns. The ‘creativity’ of pattern formation resides in the system; commonly, a large variety of qualitatively different patterns can be realized by a specific system, whereas the gradient may only be quantitatively increased (so-called phase transitions as in Figure 2, cf. the Feigenbaum scenario).

The phase transition of Figure 2 is a frequent scenario that occurs in bistable figures prominent in Gestalt psychology, such as the Necker cube (Figure 3, left). The ‘cube’ (which is actually a two-dimensional line drawing) is commonly and spontaneously perceived three-dimensionally in two versions, and most viewers experience switches between the two versions. The (psychological-motivational) gradient that causes the 3D perception is the tendency towards simplicity (*Prägnanztendenz* in Gestalt psychology [12,13]), which also explains that only one Gestalt is viewed at the same time; but why the switches between the two Gestalt patterns? It seems that the gradient that causes the formation of one pattern becomes depleted by the presence of this same pattern. It was shown that this retrograde effect—patterns gradually attenuating the control parameters, which caused their very formation in the first place—is a general phenomenon in self-organization [14] whenever control parameters are not kept constant by further environmental constraints. This is a central insight gained by our previous analysis [15]. So, the question arises: Which are the relevant control parameters in a joke?

In the application to psychology, motivational tension is an external gradient that leads to the self-organization of cognitive or behavioral patterns, which, in turn, reduce the tension. It may be claimed that this provides pattern formation with a functionality: those patterns are formed that are best capable of reducing their environmental gradients [16].

The Necker cube in Figure 3 (right) is an example of incomplete resolution of incongruity in the style of M.C. Escher’s impossible geometry, and it is also an imperfect pun [7]. The corners of this cube are designed in such a way as to not allow an unequivocal formation of a 3D pattern. The ongoing switching between the two patterns, the phase transitions, can, however, also be observed here. The ‘classical’ Necker cube in Figure 3 (left) is ‘complete’ and ‘perfect’ by this terminology.

We tested the propositions generated by the application of synergetics to phenomena of humor in an empirical dataset that was collected in the context of schizophrenia research [17]. The study was approved by the cantonal ethics committee (KEK) on the basis of written informed consent (approval KEK #249/09). A group of 56 healthy participants (32 male, 24 female; mean age 36.7 years) were reanalyzed for the goals of the present study. Participants were presented a set of up to 50 visual puns in random order on a computer (examples of stimuli in Figure 4). After each presentation of a pun, the participant was asked to rate his or her experienced funniness on a scale ranging from 1 to 6. Puns were presented in a Matlab environment, and the processing time for each pun was recorded. The level of incongruity contained in each pun was assessed by three raters on a three-point scale (0 = little incongruity, 1 = some incongruity, 2 = much incongruity). The raters also estimated how much mentalizing ability (i.e., theory of mind, ToM) was needed to fully understand the respective joke (0 = no ToM needed, 1 = some ToM needed, 2 = ToM needed). Finally, the raters grouped the puns into bistable displays (such as the Necker cube), perfect puns, and imperfect puns.

Further ratings were performed additionally for the present study by two raters. First, the puns were classified into bistable displays and puns allowing complete and incomplete incongruity resolution. Using six-point Likert scales, each pun was assessed as to how much *Schadenfreude* was involved, and the degree of nonsense each pun contained.

For example, the sketched pun in Figure 4 (top left) was rated as medium high in *Schadenfreude* (mean = 3) and high in nonsense (mean = 5). This pun was classified as incomplete, imperfect [7], and needing mentalizing ability (2 = ToM needed). Mean funniness experienced by participants was 4.21 in this pun. The Arcimboldo painting of Figure 4 elicited little *Schadenfreude* (mean = 1.5) and little nonsense (mean = 2.5), represented an incomplete pun, a perfect pun, and was assessed by participants as rather funny (mean = 3.68). Raters considered that no mentalizing ability was needed to understand the Arcimboldo pun. The incomplete and imperfect pun in Figure 4, top right, was not entered in the study, and is shown for illustration purposes.

Further data were acquired by questionnaires. The Five Factor Personality Inventory (NEO-FFI, [18]) is a self-assessment of personality dimensions (the ‘Big Five’: neuroticism, extraversion, openness for new experience, agreeableness, conscientiousness). The five dimensions are measured by 60 items with five-point scales.

The Mehrfachwahl-Wortschatz-Intelligenztest (MWT, multiple-choice vocabulary intelligence test, [19]) is a brief test to assess the passive vocabulary of a person (an aspect of general intelligence). The test consists of 37 items; each item offers a row of five words of which only one is a valid expression in German. Participants have to mark the valid word in each item.

Prior to presentation of the humorous stimuli, all participants rated their own momentary affective state using the Positive and Negative Affect Scale, PANAS [20]. It consists of twenty emotion adjectives (e.g., ‘active’, ‘interested’, ‘upset’, ‘afraid’) that are rated on five-point scales ranging from ‘very slightly or not at all’ to ‘extremely’. The twenty items load on two factors, positive affect and negative affect.

The MASC (Movie for the Assessment of Social Cognition) is a test to assess ToM in a movie depicting various social interactions [21]. The movie is interrupted several times for a multiple-choice display offering options on the thoughts or motifs of the actors at that point in the movie. The correct option can be inferred if the participant uses perspective taking and is able to understand the social cognition of the actors. The sum of all correct responses provides a measure of this participant’s mentalizing ability.

To study the association of experienced funniness, the dependent variable, with the properties of the puns and the traits or states of participants, hierarchical regression analyses were conducted using JMP Pro 15.1 (SAS Institute Inc., Cary, USA). As in the dataset, funniness assessments are nested in 56 participants and in up to 50 stimuli; the random effects Participant# and Pun# were considered when testing the associations between funniness and each fixed effect. The fixed effects were the variables put forward in the hypotheses, hence mentalizing abilities, disparagement/*Schadenfreude* content, personality traits of participants, degree of incongruity of the stimuli, duration of incongruity resolution, and type of the stimuli.

## 3. Results

### 3.1. Self-Organization Model of Humor

#### 3.1.1. Step 1: Emergence of Attractor/Gestalt Formation

The initial Gestalt or cognitive schema is formed by the information of stimulus S1 given in a verbal joke before the punchline, the point of the joke. In a visual pun, it is the overall Gestalt-like scene recognized first (for example, in Figure 4 top left, the scene is that of two men skiing downhill in the mountains). As detailed before, we may assume that the emerging pattern can be presented by an attractor A1 with a potential minimum *V*_1_. The basin b1 of attractor A1 comprises all pieces of information contained in S1. This is illustrated in Figure 5.

#### 3.1.2. Step 2: Incongruity

Then, the punchline of the joke provides further information that is contained in stimulus S2. S2 is incompatible with S1 and inserts incongruity to the system, which destroys attractor A1 together with its basin b1. Psychologically, disorientation and cognitive dissonance is experienced, and the system state is forced back to the initial condition *V*_0_. In terms of Friston’s free-energy principle [22,23], a prediction error has occurred and free energy is again available (Figure 6). The motivational tension contained in the dissonance and the newly available free energy both stimulate subsequent novel pattern formation.

Interestingly, in verbal jokes at this point the joke telling is completed, but the joke is not. The recipient is left with an open Gestalt. The situation is different in visual puns, where the stimulus remains visually accessible, not only via memory.

#### 3.1.3. Step 3: Resolution of Incongruity

The dissonance resulting from Step 2 motivates novel pattern formation in order to integrate the surprising information of stimulus S2, which constitutes the point of the joke, into a novel attractor A2 (Figure 7). If all information of S1 and S2 is covered by A2 we speak of complete resolution of incongruity, and the basin of the novel attractor is b2. If not all of the information can be accommodated by the recipient, the joke is called incomplete, and the basin b2′ of attractor A2′ is smaller. Accordingly, some incongruity is left after incomplete resolution, and there are still remainders of non-sense as these remainders cannot be integrated in the novel attractor. Comparing A2 to A1, the novel attractor A2 not only possesses a wider basin than A1, but also has a larger minimum potential *V*_2_ than the previous attractor A1. The reason for *V*_2_ > *V*_1_ is that A2, in all jokes and puns, is less probable than A1. The initial attractors A1 in humor express more common, often everyday situations and scripts, whereas A2 is unconventional, surprising, and sometimes nonsensical, and thus less likely. Generally, we may hypothesize that the amount of resolvable incongruity *V*_0 −_
*V*_1_ predicts the funniness of a joke or pun (Hypothesis 5).

One question has puzzled humor researchers [24]: Why is the resolution of incongruity so funny and joyful? Common problem solving does also resolve incongruity in the shape of conflicting information, yet problem solving is not experienced as funny. The model of Figure 7 suggests an answer: As A2 is less probable than the previous A1, there results a net gain of free energy. This surplus free energy, set free in the space of less than a second, may be the reason why jokes are joyfully appreciated and make people laugh suddenly. We therefore expected that the faster the resolution, the funnier the pun (Hypothesis 6).

In incomplete jokes, A2′ should have larger minimum potential *V*_2_ than in complete jokes because dissonance reduction is smaller owing to the remaining incongruence, thus the nonsensical content. At the same time, free-energy gain is higher, which may mean that incomplete jokes are funnier (Hypothesis 7).

The dynamics of incongruity resolution is, however, markedly different in bistable puns: Here, attractors A1 and A2 are equally likely and usually have similar potentials *V*. This is the case in the prototypical Necker cube (Figure 3) as well as in Escher-style bistable puns of this study. Bistable puns do not afford enduring incongruity resolution, which is supported by the ongoing availability of the visual stimulus. Hence the typical perception of switching between attractors, which may be surprising, but not experienced as very funny, as no dissonance reduction or free-energy gain are accomplished. Therefore, bistable puns were expected to be experienced as less funny (Hypothesis 8).

In addition to the four general hypotheses formulated in the introduction, we therefore derived Hypotheses 5 to 8 from the model described in Section 3.1.

**Hypothesis 5.** the degree of incongruity *V*_0 −_
*V*_1_ is associated with experienced funniness;

**Hypothesis 6.** faster incongruity resolution is associated with experienced funniness;

**Hypothesis 7.** incomplete puns are experienced funnier than complete puns;

**Hypothesis 8.** bistable puns are experienced as less funny.

In the next section, all hypotheses will be examined based on the empirical dataset with visual puns.

### 3.2. Empirical Study Results

#### 3.2.1. Hypothesis 1: Perceived Funniness Is Related to Mentalizing

The participants’ mentalizing abilities were assessed by the MASC test, and the visual puns were rated into three categories expressing whether mentalizing ability/theory of mind was needed to get the joke (0 = no ToM needed, 1 = some ToM needed, 2 = ToM needed). Comparisons of the mean funniness of the three groups of puns per participant were conducted in hierarchical regression analysis (mixed-effects modeling). A strong effect was found showing that ToM puns were rated significantly funnier overall (Table 1, Model 1.2), and this model had, according to Akaike’s Information Criterion (AICc), a better fit than the null model (Model 1.1). Mentalizing ability measured by the MASC test did not add to model fit and was not significant (Model 1.3). Further analyses showed that mentalizing ability predicted perceived funniness of puns only in category 0 (no ToM needed), where a negative relationship with funniness was found (*t*(55)=−2.47, *p* = 0.017). Thus, when the jokes did not address mentalization, participants with higher mentalizing ability rated them less funny.

The *Schadenfreude* ratings of all puns (raters’ assessments) were significantly associated with their mean funniness assessed by participants (Model 1.4).

Thus, results supported that mentalization-related puns and puns rated to possibly elicit *Schadenfreude* were experienced funnier, but this effect was not predicted by participants’ individual mentalizing abilities.

#### 3.2.2. Hypothesis 2: Perceived Funniness Is Related to Affective States

Positive and negative affect was measured using the PANAS in all participants prior to their being presented the humor stimuli. We tested how affect influenced all aggregated funniness ratings of each participant. Positive affect was significantly associated with higher subsequent mean funniness ratings (*t*(55) = 2.64, *p* = 0.011); negative affect was not significantly associated (*t*(55) = 1.94, *p* = 0.058).

#### 3.2.3. Hypothesis 3: Perceived Funniness Is Related to Personality Traits

The “Big-Five” personality factors were assessed using the NEO-FFI, and these factors were used to predict each participant’s mean funniness ratings. In a multiple regression model, funniness was not significantly predicted by any of the factors, and the whole-model test was not significant (*F*(5,51) = 1.63, *p* = 0.17).

#### 3.2.4. Hypothesis 4: Perceived Funniness Is Related to Intelligence

The MWT test allowed for the estimating of an aspect of general intelligence of study participants. The association of this measure with mean experienced funniness was not significant (*t*(54)=−0.28, *p* = 0.78).

#### 3.2.5. Hypothesis 5: Perceived Funniness Is Related to the Degree of Incongruity

Incongruity of each pun was rated by investigators, and funniness, the dependent variable, was rated by each participant per each pun. The dependent variable was thus hierarchically nested in Participant# and in Pun#, the two random effects of a mixed effects regression model. Table 2 presents the results of the null model (Model 2.1). Model 2.2 suggested that incongruity is a significant predictor of funniness. The information criterion AICc points to Model 2.2 being superior to the null model.

#### 3.2.6. Hypothesis 6: Perceived Funniness Is Higher with Faster Incongruity Resolution

The timing of incongruity resolution of a pun was estimated using, as a proxy variable, the time in seconds needed by participants for processing the information of a pun stimulus, including their decision on and rating of its funniness. This was computed by the duration after which participants freely switched to the next pun. The mixed effects model (as in Table 2) of the dataset, however, did not converge, so the data of all ratings and durations were aggregated per each of the 50 puns. A *t*-test of the aggregated data showed that mean pun funniness (the dependent variable) and mean pun duration (the predictor) were negatively associated (*t*(48) = –2.42, *p* = 0.02). This means that funnier puns were processed faster, consistent with Hypothesis 6.

#### 3.2.7. Hypothesis 7: Perceived Funniness Is Higher in Incomplete and Imperfect Puns

The predictor ‘Complete/Incomplete/Bistable’ in Table 3 is a categorial variable with three steps according to the classification of puns into these three categories. We found that bistable puns were rated the least funny, and incomplete puns as the funniest, which would be in line with the hypothesis (Model 3.1). Resolution of the incongruity of a pun, however, can be assessed only in puns that tell a narrative and have a script. The concept of complete or incomplete incongruity resolution cannot be meaningfully applied to scriptless bistable puns. We, therefore, studied completeness without the bistable puns in a smaller sample of *n* = 1759 puns. The distinction ‘complete’ versus ‘incomplete’, with all bistable puns excluded, was no longer statistically significant (Model 3.2). We repeated this procedure with the assessment of puns being classified ‘perfect/imperfect/bistable’. Here, both Models 3.3 and 3.4 showed that funniness was associated with this categorial variable, also with bistable stimuli excluded (*n* = 1759). The raters’ assessments of the nonsense contained in all puns was used as a fixed effect in Models 3.5 and 3.6 of Table 3. Nonsense was rated funnier overall, yet again, when bistable puns were excluded, the variable nonsense no longer predicted participants’ experienced funniness. Thus, Hypothesis 7 was supported only insofar as imperfect puns were experienced as funnier than perfect as well as bistable puns.

#### 3.2.8. Hypothesis 8: Bistable Puns Are Experienced as Less Funny

The funniness of bistable puns was compared to all other puns using the participants’ mean funniness ratings per pun. Under the assumption of unequal variances of the two groups, funniness of bistable stimuli was significantly lower (*t*(46.5)= 8.47, *p* < 0.0001), in support of Hypothesis 8. On the scale ranging from 1 to 6, grand mean funniness of the *n* = 16 bistable puns was 2.37, and of the other 34 puns, 3.57.

## 4. Discussion

The model of humor developed in this article rests on the premises of complexity theory, specifically the self-organization theory elaborated over the past decades by Haken’s synergetics. For the context of humor appreciation, we assumed that the mind constitutes a complex open system, which shows emergent pattern formation under the influence of external constraints, the control parameters. These parameters are commonly energy gradients in physical systems, and in the application of synergetics to mental and information-processing systems, they have the shape of motivational forces and affordances [9,10,15]. In the terminology of the free-energy principle, free-energy minimization is the equivalent of a system relaxing to a potential minimum (K. Friston, private communication; [23]), and free energy acts as a control parameter for the mind.

In our modeling approach, we focused on the treatment of verbal jokes and visual puns as prototypes of the wider field of humor. In accordance with the majority of humor research, we assumed that the experience of funniness, hilarity, and mirth that follows from humorous stimulation is a result of the incongruity between components of the stimulus; the recipient of the joke or pun must resolve this incongruity ‘to get the joke’. On the background of complexity theory, this dynamic constitutes a phase transition: a subset of the stimuli provokes an initial pattern formation process, which is then critically contradicted by the punchline of a verbal joke, or by further material embedded in a visual pun. This incongruity or contradiction is resolved by new pattern formation that mends and integrates, sometimes only incompletely, the incongruent stimuli. We represented the core process of phase transition by the potential differences between the initial pattern, the state after destruction of this initial pattern, and the resolution pattern. In psychological terminology, these potential differences equate to cognitive dissonance [25], a motivational parameter with negative valence, and their resolution to dissonance reduction, and thus to negative reinforcement. In the view of the free-energy principle [22], the potential difference entailed by incongruity means that free energy becomes available to the recipient of the joke.

We derived several hypotheses from this model, which we tested in an empirical dataset of responses to visual puns displayed to a sample of participants. We found that the degree of incongruity and the speed of incongruity resolution contributed to experienced funniness in accordance with the hypotheses. Bistable puns were assessed as clearly less funny because, again consistent with the model, there is little or no potential difference between the subsequent patterns in bistability, and resolution is not attained owing to ongoing phase transitions when viewers continue switching back and forth. Further assumptions based on the model were not clearly supported; this was true for the hypothesis that incomplete resolution (so-called nonsense jokes) may contribute to funniness. In the present database, this effect was confounded by the influence of the bistable stimuli, and should be tested in further specifically designed experiments.

We additionally formulated exploratory hypotheses on the basis of previous research on humor. Empirical data supported the expectation that disparagement or *Schadenfreude* contents in the puns should contribute to funniness, and it was also true that social mentalization-based content of puns was experienced as funnier. Prior positive affect, but not personality traits and intelligence, was significantly associated with responses to the puns. Interestingly, prior negative affect pointed in the same direction as positive affect, but failed to be significant.

In general, we believe that the present model of humor successfully integrates known phenomena related to humor and can easily accommodate the mentalizing and *Schadenfreude* findings: at the decisive moment of fast incongruity resolution, further motivational forces linked to semantic social factors can amplify the drivers (free energy and dissonance) during the formation of the final resolution pattern. This enlarged view is consistent with the formal model of a phase transition as the core of humor experiences.

Limitations of the present modeling approach may lie in the overly simplified representation of the attractors A1 and A2 arising from the presentation of humorous stimuli. First, it is plausible that instead of the simple point attractors depicted in Figure 5, Figure 6 and Figure 7, more complicated attractor landscapes may result. Phase transition dynamics in this case would then have more degrees of freedom, and the essential phase transition and incongruity resolution is not sufficiently described by the difference of *V*_0_, *V*_1_, and *V*_2_. Furthermore, a general objection may be raised, as the model is based on continuous attractors emerging from the response to semantic informational inputs, but narrative stimuli in a joke do not constitute a differentiable continuous variable but categorial data. A way to avoid the problems of categorial versus continuous data is to adopt the second foundation of synergetics [10], which is based on probability distributions, and thus continuous entities instead of (categorial) pieces of semantic information. Probability distributions are also the basis of the statistical analysis of the empirical data on visual puns.

As an outlook, the scope of the present model may be opened up to apply to mental change dynamics outside of humor. We mentioned above that there is the open question why simple problem solving is not funny. In the perspective of our model (cf. Figure 7), the reason may be that the new pattern, attractor A2, in the case of a real-world solution, is not less probable than the problem pattern A1, so that A2 possesses a lower potential. No free-energy gain should then result from the solution, and hence little surprise and funniness.

It appears promising to also apply the model of a mental phase transition generally to decision processes in the context of volitional psychology [26]. The core issue in volition is action control—how to ‘cross the Rubicon’ between mere cognitive intentions on the one bank of the Rubicon towards goal-directed overt behavior on the other. Intuitions derived from the phase-transition model may lead to testable hypotheses on how decision making can be alleviated.

A further application is viewing change processes in psychotherapy on the background of self-organizational dynamics; several of the so-called common factors of psychotherapy [27] are likely related to contextual gradients that were hypothesized to induce phase transitions in clients’ behavior and cognition [11]. According to this approach, mechanisms of change may be distinguished into deterministic, stochastic, and phase-transition interventions. Designing the contextual gradients such that incongruity resolution occurs would support therapeutic changes—a possible transfer from the dynamics of humor to psychotherapy process.

## Figures and Tables

**Figure 1 entropy-25-00341-f001:**
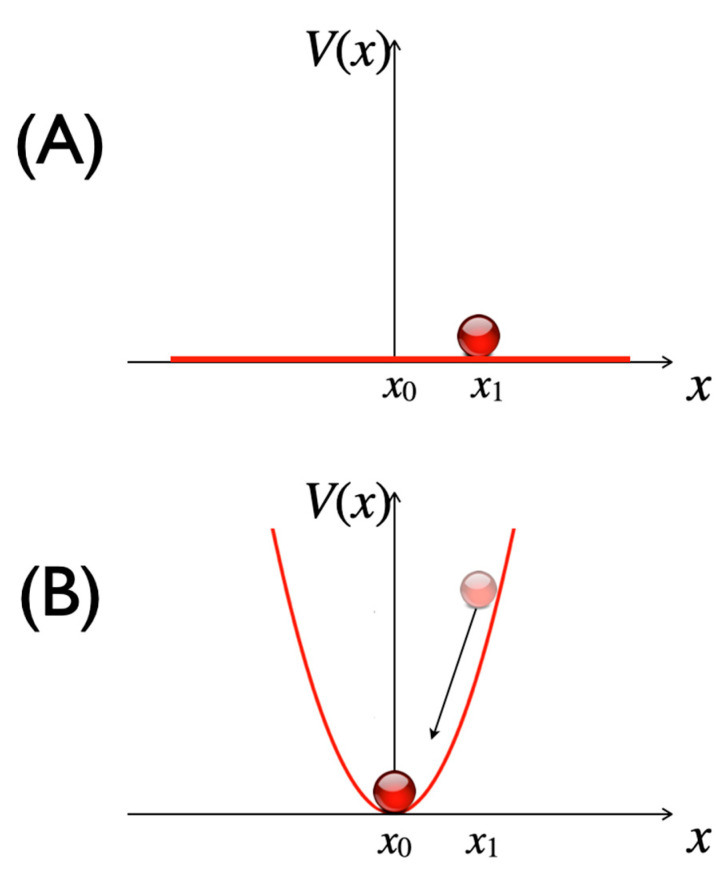
In condition (**A**), all states *x* (e.g., *x*_1_) of a system have the same probability. After self-organization, complexity is reduced to few (here, one) order parameters in (**B**). An attractor has evolved that can be described by a potential function *V*(*x*) with a minimum *x*_0_. *V*(*x*) is represented by the red graph, the system state by the red circle. States outside the attractor, such as *x*_1_, are deterministically drawn into the potential minimum.

**Figure 2 entropy-25-00341-f002:**
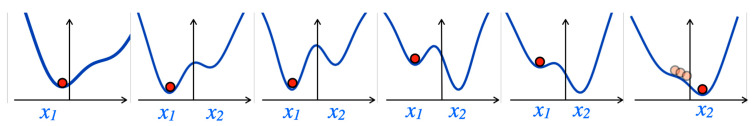
Illustration of a phase transition: The attractor landscape gradually changes (from left to right) due to changes of the environmental gradient. Respective system state symbolized by red circles. The attractor at *x*_1_ is destabilized and gives way to a novel attractor at *x*_2_.

**Figure 3 entropy-25-00341-f003:**
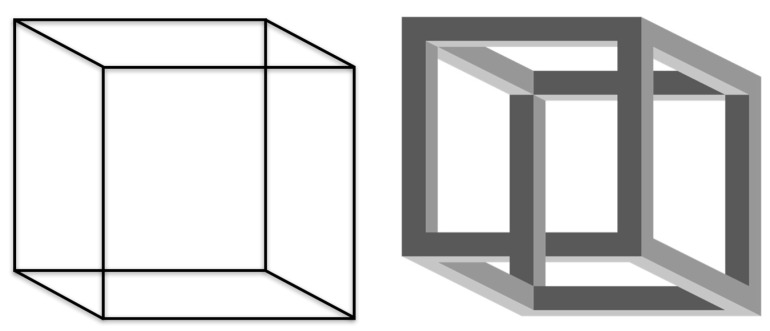
**Left**, the Necker cube. **Right** (GNU licence Wikipedia), the impossible Necker cube.

**Figure 4 entropy-25-00341-f004:**
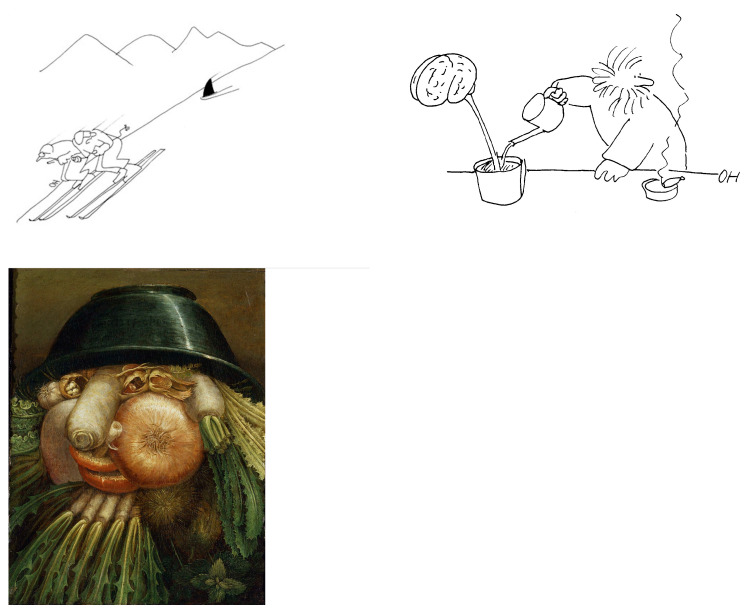
Examples of visual puns. Top: Two drawings, courtesy of Oswald Huber ©. Bottom: “The Vegetable Gardener”, painting by Giuseppe Arcimboldo (public domain).

**Figure 5 entropy-25-00341-f005:**
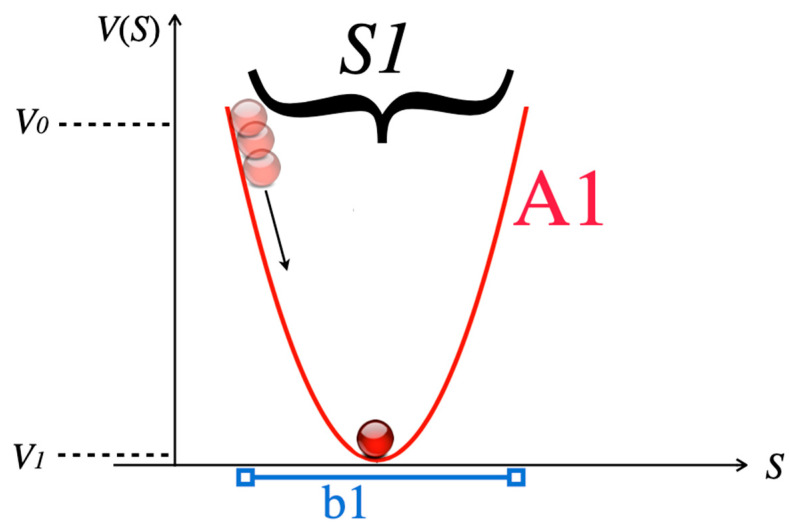
The stimulus S1 provided by the joke prior to the punchline creates a cognitive pattern with accompanying attractor A1. *V*(S) is the potential of the stimuli given. The initial condition of the system (red circles) is at *V*_0_, and the system relaxes to *V*_1_ as soon as the information in S1 has been provided completely.

**Figure 6 entropy-25-00341-f006:**
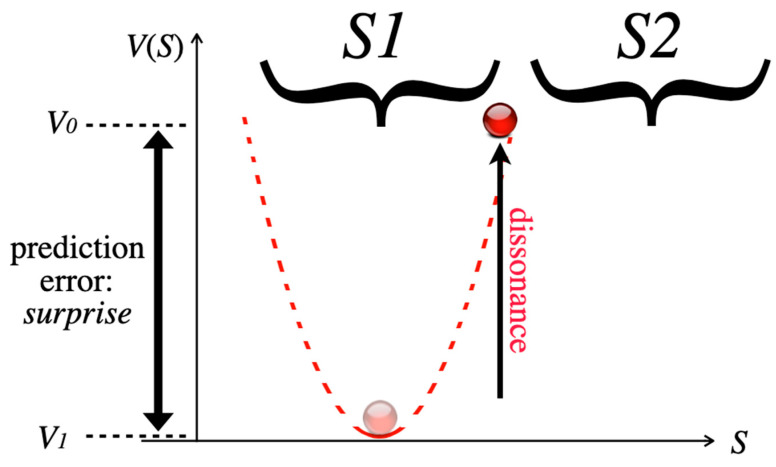
The punchline/point of the joke in stimulus S2 is incongruous with S1, so that attractor A1 can no longer be maintained. The system (red circle) is forced back to its initial condition at *V*_0_.

**Figure 7 entropy-25-00341-f007:**
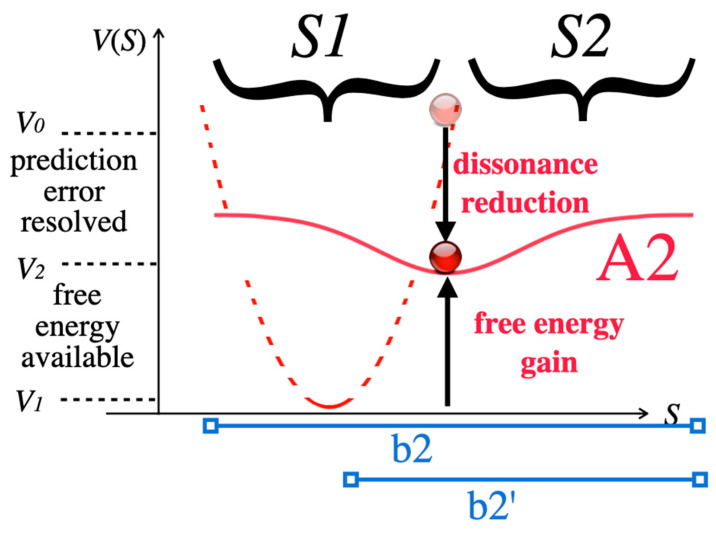
The information of S1 and S2 is integrated by novel attractor A2, which resolves the incongruity. The basin of A2 in complete jokes is b2 as A2 integrates all information S1 + S2. In incomplete jokes, the basin b2′ of an attractor A2′ (not shown) is smaller. The system state (red circle) relaxes to potential *V*_2_.

**Table 1 entropy-25-00341-t001:** Mixed effects models of *N* = 56 participants each viewing between 40 and 50 visual puns. Dependent variable, mean funniness of a pun per participant and category (no ToM needed/some ToM needed/ToM needed).

	Model 1.1 (*n* = 168)	Model 1.2 (*n* = 168)	Model 1.3 (*n* = 168)	Model 1.4 (*n* = 168)
Fixed Effects:				
ToM needed?	–	*F*= 91.7 ****	*F*= 91.7 ****	*–*
MASC	*–*	*–*	*F*= 3.51	*–*
*Schadenfreude*	–	–	*–*	*F*= 180.2 ****
Random Effects:				
Participant# (% variance)	34.7	65.9	64.7	65.4
*r*^2^ (% total variance)	47.6	82.3	82.3	82.0
AICc	459.2	361.9	365.2	357.2

Note. AICc, Akaike’s Information Criterion corrected; ToM, theory of mind; MASC, Movie for the Assessment of Social Cognition (mentalizing ability); *n*, number of observations; **** *p* < 0.0001.

**Table 2 entropy-25-00341-t002:** Mixed effects models of *N* = 56 participants each viewing between 40 and 50 visual puns. Dependent variable, funniness of a pun.

	Model 2.1 (*n* = 2509)	Model 2.2 (*n* = 2509)
Fixed Effect:		
Incongruity	–	*t* = 6.0 ****
Random Effects:		
Participant# (% variance)	21.53	24.13
Pun# (% variance)	25.05	15.96
*r*^2^ (% total variance)	48.0	48.0
AICc	7996.9	7974.1

Note. AICc = Akaike’s Information Criterion corrected. *n*, number of observations; **** *p* < 0.0001.

**Table 3 entropy-25-00341-t003:** Mixed effects models of *N* = 56 participants each viewing between 40 and 50 visual puns. Dependent variable, funniness of a pun.

	Model 3.1 (*n* = 2509)	Model 3.2 (*n* = 1759)	Model 3.3 (*n* = 2509)	Model 3.4 (*n* = 1759)	Model 3.5 (*n* = 2509)	Model 3.6 (*n* = 1759)
Fixed Effects:						
Complete/Incomplete/Bistable	*F* = 22.8 ****	*t*= –1.0	*–*	*–*	–	–
Perfect/Imperfect/Bistable			*F* = 33.6 ****	*t* = 2.87 **		
Nonsense	–	–	–	–	*F*= 10.4 **	*t*= –0.69
Random Effects:						
Participant# (% variance)	24.6	25.2	25.3	26.0	22.4	25.1
Pun# (% variance)	14.5	16.8	11.8	14.0	21.8	17.1
*r*^2^ (% total variance)	48.0	44.4	47.9	44.3	48.0	44.4
AICc	7972.8	5724.5	7963.1	5718.3	7993.2	5726.1

Note. AICc = Akaike’s Information Criterion corrected. *n*, number of observations; ** *p* < 0.01; **** *p* < 0.0001.

## Data Availability

No new data were created. Data are unavailable due to privacy restrictions warranted in the written informed consent.

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
