# Peer review of "A Complexity Science Account of Humor"

_entropy, 2023, doi:10.3390/e25020341_

Round 1
Reviewer 1 Report
In this study, the Authors tested the propositions generated by the application of synergetics to phenomena of humor in an empirical dataset that was collected in the context of schizophrenia research. They suggested that for the context of humor appreciation, the mind constitutes a complex open system, which shows emergent pattern formation under the influence of external constraints, the control parameters. Using multisided methodical approach, the Authors obtained a lot of interesting results and created several hypothesis. The Authors also suggest that the obtained results can be applied to mental change dynamics outside of humor.
This article is provides interesting results and can be published.
There are several small concerns...
1 The text is too verbose and in some place difficult for perception.
For example, this fragment of Methods containing lines 189-201
“A group of 56 healthy participants (32 male, 24 female; mean age 36.7 y.) were included as a control group in the original study; these participants were reanalyzed for the goals of the present study. Participants were presented a set of 50 (after initial 33 participants reduced to 40) visual puns in random order on a computer (examples of stimuli in Figure 4). After each presentation of a pun, the participant was asked to rate his or her experienced funniness on a scale ranging from 1 to 6 ("How funny do you think this picture is?"). Puns were presented in a Matlab environment, and the processing time for each pun was measured. The level of incongru- ity contained in each pun was assessed by three raters on a three-point scale (0=little incongruity, 1=some incongruity, 2=much incongruity). The raters also estimated how much mentalizing ability (i.e., Theory of Mind, ToM) was needed to fully understand the respective joke (0=no ToM needed, 1=some ToM needed, 2=ToM needed). Finally, the raters grouped the puns into bistable displays (such as the Necker cube), perfect puns and imperfect puns”.
2 You perfectly outlined potential applications of your study in conclusion. Please, insert the main of them into the Abstract.
3 In this article the Authors provided new data of the investigation of humor processes using a model of self -organization. Based on the obtained results, they concluded that this model may be opened up to apply to mental change dynamics outside of humor, particularly to problem solving and goal-directed overt behavior.
Therefore, this article contributes to a new level of understanding of human cognitive processes which confers it a special scientific soundness.

Author Response
Please see the attachment
(changes in the manuscript R1 are color-coded yellow)

Reviewer 2 Report
The authors provide an original model of funniness perception based on free energy minimization and synergetics. Predicted results are supported by a mixed model regression. I think the paper is suitable for publication and I really hope to see it published soon. Please find minor and major comments here below.
Majors:
After reading the introduction I felt I was missing the relation between the authors’ working hypothesis and physiological and behavioural correlates of funniness (such as the ones mentioned in the last few lines of page 1): what are the relation between incongruity resolution and arousal, laughter, pleasure, muscle contractions, attention and so on. I think the introduction would benefit from an additional low-level of description linking incongruity and physiology of funniness.
Take the example of the Necker cube. It satisfies your incongruity hypothesis (there is a novel Gestalt disrupting the previously expected one) but still it is not funny. It is not completely clear what is the difference between the cube and one of the jokes you use an example in your introduction. I would better explain, as soon as in the introduction, what are the peculiar social elements that make jokes funny and that are not involved in the perception of ambiguous images as the cube. This might relate to to what you briefly mention here: “To get the joke in social and 47 especially disparagement humor, the perceiver must switch between the perspectives of 48 the protagonists in the joke or pun, and thus must possess mentalizing abilities, or in cog- 49 nitive terminology, Theory of Mind (ToM)”. Still, I think it deserves more lines. I was asking myself: do incongruent perceptual schemes (e.g. stimulus S1 and S2 in the boss/student joke) need to be of a special kind to make jokes funny?
The paper has a strange organization into sections. For example it is weird to find the three steps of humour (described in § 3.1. Self-organization model of humor) and the corresponding expected results (hypotheses 5,6,7 and 8) in the result section. Is it because of word limit of the introduction? In this case maybe the editorial office could allow for a more flexible word limit for the benefit of readability. The detailed description of the model and its steps really helped me to understand the general aims and design better. I is a pity not to have it in the introduction.
Minors:
p.2, line 65: “Schadenfreude”. Is it that different from mockery or teasing? Anyone who does not speak german will need to google it.
p.2, line 66: “Incongruity is resolved completely in this joke”. I might be wrong but I feel no difference between this joke and the Chuck Norris one. In the boss/student joke I feel some incongruity and absurdity remains: for instance the boss punchline is incongruous with the situation and absurd (in its more etimological sense “out of tune”) given the situation and common politeness. It feels no less absurd than Chuck’s superpowers. Please add some explanation on we you think one is incomplete and the other is complete. (+is the distinction between the two so important at this initial lines in the introduction?)
p.3., lines 83-86: “Visual puns can be 'per- 83 fect' [7] or 'imperfect;' they are called perfect when the ambiguous visual element is de- 84 picted in a way that is fully compatible with both scripts. Thus, perfect puns are similar 85 to jokes that allow complete resolution. Imperfect puns are the visual analogues to incom- 86 plete jokes”. Same as above is it not easy to understand incompleteness/perfectness of jokes/puns for non experts. Meanwhile I still have the impression the distinction is not so important in the introduction. Readability would benefit from some additional explaining in case the authors want to keep the distinction (e.g. I understood the difference very clearly when you showed the Necker’s cube example).
p.2, line 74: “The psychological extra reward of this and other Chuck Norris puns possibly 74 rests in the narcissistic identification with invincible Chuck”. I would either better explain this concept or eliminate it.
p.6, line 245: I would include the full list of fixed factors here. It helps the reader to keep in mind all the factors that you included in the model
table.2. r 2 (% variance) in table 2 means marginal r2, i.e. the percentage explained by fixed factors only? Or conditional, the total variance explained by the model?
p.12 line 426-428: “These parameters in physical systems are commonly energy gradients, and in the application of synergetics to mental and information-processing systems they have the shape of motivational forces and affordances”. There is quite a jump from an energetic (“energy gradients”) to a phenomenological/behavioural (motivational forces and affordances) level of description. Motivation and affordances are just the epiphenomenon of some energetic gradients that must be involved in mental self-organization. So I would try to define what is the energy/information involved in the mental world. [All through your paper you seem to suggest that it is free energy (surprisal, prediction error, entropy) but you do not clearly state it. For example you say that it is free energy that becomes available before phase transition and that potential differences are differences in free energy. Somewhere else you say that self-organization selects the patterns that are more efficient in reducing gradients, in FEP this would translate in minimizing free energy]. I don’t fully understand what prevents you from clearly saying that it is free energy gradients that motivate cognition and act as a control paramenter for the mind
Author Response
Please see the attachment. Thanks to Reviewer 2!
